# Mixed-methods study on pharmacies as contraception providers to Kenyan young people: who uses them and why?

Lianne Gonsalves ,[1,2,3] Kaspar Wyss,[2,3] Jenny A Cresswell ,[1]
Michael Waithaka ,[4] Peter Gichangi ,[4,5,6] Adriane Martin Hilber[2,3]

Correspondence to
Lianne Gonsalves;
gonsalvesl@who.int

## ABSTRACT

**Objectives** This study sought to answer two questions: (1) what are the characteristics of young Kenyans aged 18–24 who use contraception obtained at pharmacies, and (2) why are pharmacies appealing sources of contraception?

**Design and setting** This was a mixed-methods study in one peri-urban part of Kwale County, Kenya. Methods included cross-sectional survey (n=740), six focus group discussions, 18 in-depth interviews and 25 key-informant interviews. Quantitative data analysis identified factors pushing young people to pharmacies for modern contraception versus other sources. Qualitative data analysis identified reasons pharmacies were perceived to be appealing to young clients.

**Participants** Participants were (1) young people aged 18–24 from the study area, including a subset who had recently purchased contraception from a pharmacy; or (2) pharmacy personnel and pharmacy stakeholders.

**Results** Among surveyed participants who had ever had sexual intercourse and had used modern contraception at last sexual intercourse, 59% obtained it from a pharmacy. In multivariable analysis, participants who used a condom or emergency contraception as well as those living alone were significantly more likely to get contraception from pharmacies. Pharmacies were valued for their convenience, privacy, non-judgmental and personable staff, service speed, as well as predictable and affordable prices.

**Conclusions** Our findings indicate a high percentage of young people in Coastal Kenya use pharmacies for contraception. Our inclusion of emergency contraception users partially explains this. Pharmacies were perceived to be everything that health facilities are not: fast, private and non-limiting. Policy-makers should recognise the role of pharmacies as contraception providers and look for opportunities to link pharmacies to the public health system. This would create a network of accessible and appealing contraception services for young people.

## INTRODUCTION

Young people need access to contraception. However, around the world, and in low-income and middle-income countries in particular, public sector contraceptive services are not meeting this need. Data from 61 low-income and middle-income countries estimated that 33 million young women aged

## STRENGTHS AND LIMITATIONS OF THIS STUDY

⇒ Participants were asked to specify where they or their partner had obtained the contraception used at last sexual intercourse. This is a standard question for studies looking to establish contraception prevalence. However, our not further ascertaining who specifically obtained the contraception affected our ability to distinguish differences in preferences of young men versus young women.

⇒ One participant group (young people who had recently purchased contraception from a pharmacy) was recruited from five purposively selected pharmacies: this may limit the generalisability of the findings.

⇒ This study is strengthened by its mixed-methods design and inclusion of both pharmacy personnel and young people to triangulate research findings on a sensitive subject.

15–24 had an unmet need for family planning.[1] Adolescents (ages 10–19 years) and youth (15–24 years) are often reluctant to access contraception at public health facilities where they may encounter a lack of privacy, biased providers and limited contraceptive options, in addition to broader financial, legal, social and cultural barriers.[2 3]

Other parts of the health system may be able to step in to help fill this gap. In Kenya (where this study took place) and in the region, private pharmacies have become a source of modern contraception for young people.[4–7] Additional research has indicated that when contraception is introduced in pharmacies, access improves for young people.[8 9] An analysis of 33 sub-Saharan African countries found that commercial drug sellers, including pharmacies, were the source of the most recent contraceptive method for nearly one in five young people between 15 and 24 years of age.[8] When also factoring in other informal and non-medical providers, including shops, these sources together serviced nearly half of women age 15–19.[8]

Kenya's National Family Planning Guidelines allow for the provision of several kinds of modern methods[10] of contraception to be dispensed by pharmacists or pharmaceutical technologists[11] (colloquially referred to as 'chemists'). These include barrier methods like male and female condoms, as well as short-acting methods including emergency contraception (ECP), oral contraceptive pills and injectable contraception. These permissions mean that outside of health facilities, private retail pharmacies have the largest selection of modern methods available (shopkeepers can also sell condoms, per the guidelines). Private retail pharmacies must be opened by and should always operate under the supervision of either a pharmacist or pharmaceutical technologist.[12]

Despite their demonstrable popularity among young people, there are little data on the individual-level circumstances or characteristics of young people that would drive them to pharmacies for contraception. Therefore, we conducted a mixed-methods study describing how young people (aged 18–24) in Kwale County obtain contraception from pharmacies. Kwale County is one of six counties in Kenya's former Coast region. Young people between the ages of 15 and 24 were projected to make up 19% of the county's population by 2018.[13] In 2014, contraception prevalence in the county was 38%, lower than the national average of 53%.[14]

In this analysis, we sought to answer two questions: (1) what are the characteristics of young people who use contraception obtained at pharmacies, and (2) why are pharmacies appealing sources of contraception to young people?

## METHODS

The study took place in the peri-urban areas of Kwale Town and Ukunda, as well as the stretch of highway connecting the two towns. Data collection took place between October 2017 and March 2018. We used several methods (captured in table 1) to understand the experiences of pharmacy personnel and young people themselves. This study was partly nested in the ARMADILLO randomised controlled trial (RCT),[15] which assessed the effect of an unrelated digital health intervention on sexual and reproductive health–related outcomes for young people aged 18–24.

To capture the perspectives of young people, a cross-sectional survey of young people age 18–24 captured demographic information and contraceptive use patterns, including source of last contraception (these questions were one section of a broader survey conducted as part of the baseline assessment for the ARMADILLO trial). The sample size was calculated based on the ARMADILLO

**Table 1** Study methods

| Method | N | Eligibility criteria | Relevant topics addressed |
|---|---|---|---|
| Cross-sectional survey* | 740 | ► Age 18–24 | ► Contraception used at last sexual intercourse and source |
| | | ► Literate | ► Demographic and behavioural characteristics |
| | | ► Have their own mobile phone (with them at time of recruitment) and report regular use | |
| | | ► Report current use of text messaging | |
| Focus group discussions* | 6 | ► Age 18–24 | ► Sources of contraception for young people |
| | (58 participants) | ► Community members | ► Characteristics of young people who use each source |
| In-depth interviews | 18 | ► Age 18–24 | ► Reasons for having purchased contraception from pharmacy |
| | | ► Recently purchased contraception at pharmacy | ► What was valued (and not valued) about experience |
| Key-informant interviews | | ► Age 18+ | ► Characteristics of young people who purchase contraception |
| | 19 (pharmacy personnel) | ► Pharmacy personnel (any role) OR | ► What clients appreciate about experience |
| | 6 (stakeholders) | ► Pharmacy-related stakeholder (Ministry of Health; regulatory agency; professional association; non-governmental organisation) | |

*Methods which were nested in the broader ARMADILLO study, a digital health intervention randomised controlled trial. Inclusion/exclusion criteria for these nested methods were determined by ARMADILLO's objectives.

trial's primary outcome—the full protocol for the trial has been previously published,[15] along with details of participants recruited.[16]

To identify participants, we obtained a map of the study area from the Kenya National Bureau of Statistics. The KNBS divides the country into so-called enumeration areas (EAs) in preparation for the country's 2019 census. EAs consist of blocks of households. Each EA had approximately 100 households. In October 2017, data collectors enumerated all age-eligible young people in every household using a random selection of 21 EAs in the study area. From this list of age-eligible youth, a random selection of households and random selection of one youth per household was generated. Data collectors visited the selected households to recruit participants (who met eligibility criteria captured in table 1) starting in February 2018.

In addition, six focus group discussions (FGDs) were conducted with young people aged 18–24, purposively recruited from the community by data collectors. Finally, we conducted in-depth interviews (IDIs) with 18 young people aged 18–24 who had recently purchased contraception from pharmacies. We purposively recruited these young participants in one of two ways. First, we stationed a young data collector outside of well-trafficked pharmacies over three evenings, who recruited young people purchasing contraception. Second, several pharmacists in the study area were provided with leaflets with study information and requested to provide these to young contraception purchasers at the end of a transaction.

To capture the perspectives of pharmacy personnel, data collectors mapped all private, retail pharmacies in the study area using a digital form with an embedded geolocator. A random subset of pharmacies was generated using the random number generator in Excel. Pharmacies were well distributed across the study area. In each selected pharmacy, data collectors were instructed to approach the first person behind the counter, regardless of rank or level of training, explain the study and ask if they would be interested in participating. Nineteen interviews in total were conducted. An additional six key-informant interviews were conducted with stakeholders from the regulatory Pharmacy and Poisons Board, Ministry of Health, professional associations and non-governmental organisations. These were conducted in the individuals' offices in either Ukunda, Mombasa or Nairobi. Stakeholder participants were contacted first by phone or email, the study explained and a convenient time for an in-person visit set.

### Data collection and management
We obtained informed consent from all participants prior to participation. All data were collected in English, Swahili or a mix of the two, depending on participants' preference. Quantitative surveys were close ended and administered using webforms on a tablet. Data collectors entered responses save for the questions related to participants' sexual and contraceptive use history; here, to reduce potential discomfort and response bias,

participants entered their own responses. Interviews and FGDs used semi-structured guides: FGD (online supplementary file S1), IDI (online supplementary file S2) and key-informant interview (online supplementary file S3) guides are provided as online supplementary material, as are relevant survey components (online supplementary file S4). Qualitative data collection was informed by ground theory,[17] allowing us to adopt an iterative approach, with question guides modified based on emerging themes. Qualitative data collection ceased on reaching saturation. All qualitative methods used audio-recording (with participant permission). All study activities were conducted in a private location. Data collectors, speaking both English and Swahili, were recruited from the study area and specifically trained for this study.

### Patient and public involvement
Our population (young people) were directly involved in parts of the study's design and implementation. Our survey data collection team consisted of young people recruited from the study area (Kwale County). Qualitative method data collectors were also young people recruited from both Kwale and Mombasa Counties. We relied on their insight and lived experience to determine how young people would feel most comfortable being recruited. We jointly designed our recruitment and consenting procedures. A dissemination meeting involving local, county and national stakeholders (including some pharmacy stakeholder participants) took place in June 2019. Several young data collectors were invited to attend and they provided commentary on the findings.

### Researcher characteristics and reflexivity
Data collectors were young people (nearly even numbers of men and women—24 in total) recruited from Kwale and Mombasa counties. Kwale County data collectors were familiar with the study area and recognised within their communities, which facilitated enumerating pharmacies, recruiting youth participants and getting consent to interview pharmacy personnel. They were also less educated and less experienced than data collectors from Mombasa County. This, at times, resulted in a subordinate dynamic with some pharmacy personnel participants who were university educated. The first author conducted all interviews with pharmacy stakeholders. She is from the USA (from a racial minority group different from the study population) and presented as an outsider (someone not from Kenya) to interviewees. Her position (leading the study and professional affiliations) resulted in respondents treating her collegially and being open to participate.

### Analysis
Quantitative data were analysed in Stata V.14. The subject of the analyses (as described in figure 1) were survey participants who reported using one of four contraception commodities available in pharmacies (either male or female condom, ECP, daily contraceptive pills

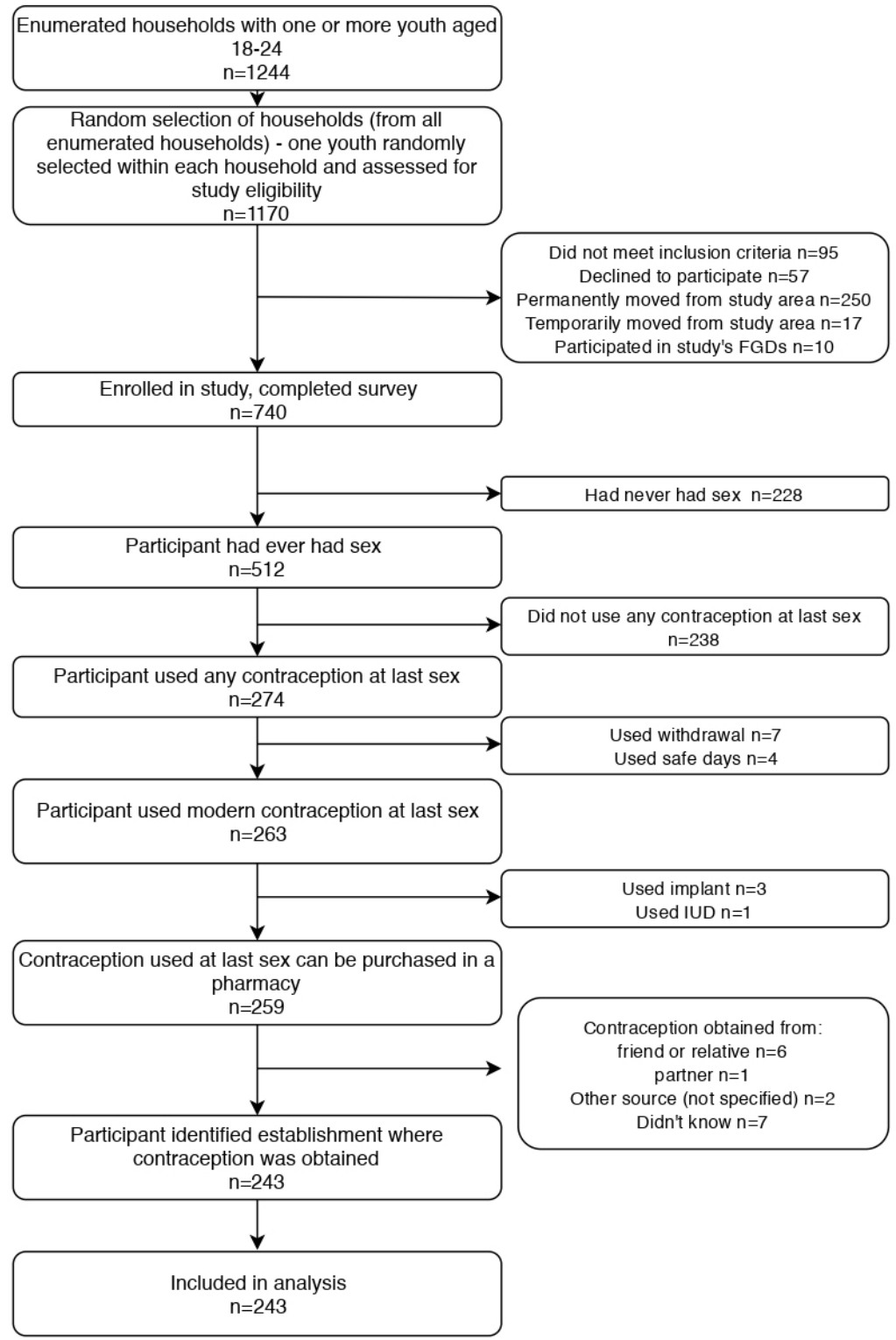

**Figure 1** Flow diagram of study participants. FGD, focus group discussion; IUD, intrauterine device.

or injectable contraception) at last sexual intercourse *and* who reported their source. Sexual intercourse was presumed to be penetrative vaginal sex. Excluded were those participants who had not used contraception at last sexual intercourse, who had not used a contraceptive commodity (withdrawal method, calendar days), who could not remember where they had obtained their

method and/or who had obtained it from a partner or friend. We developed a dichotomous 'source of family planning' outcome, distinguishing between 'pharmacy' and 'any other source'. The latter included any public or private health facility, community-based distributors, non-governmental organisations, shops, schools and supermarkets. Following descriptive statistics, bivariate

log binomial regressions assessed the association between the outcome and each behavioural/sociodemographic variable of interest. Any analysis showing a p value <0.2 moved the variable into a multivariable Poisson regression model with robust 95% CIs.

All qualitative data were analysed using the Framework Method.[18] Data were first transcribed verbatim and then translated (if necessary) into English. For a subsection of Swahili-language interviews, English-language transcripts were compared against the original Swahili-language interview audio file by another member of the research team to ensure consistency. Qualitative analysis for the broader study was guided by the five, *WHO-defined dimensions of quality health services to adolescents*: equity, accessibility, acceptability, appropriateness and effectiveness.[19] All transcripts were read once to improve familiarity with the data. Then, qualitative analysis was conducted in Atlas.ti V.8, with deductive and then inductive coding of a subset of transcripts to develop and refine a coding framework. Deductive coding was informed by the 'accessibility' and 'acceptability' dimensions and broadly captured any reference to pharmacies being 'appealing'. Inductive coding of these data then identified specific reasons for appeal, subsequently grouping these into broad categories related to pharmacy outlet, personnel and service appeal. These broad categories and individual reasons structure the presentation of the qualitative results.

## RESULTS
### Survey sample characteristics
A total of 1170 youth were approached for participation, of which 740 (63%) consented to participate and completed the survey. Reasons for non-participation are captured in figure 1. As seen in table 2, of the 740 young people aged 18–24 who participated in the cross-sectional survey, 512 (69%) had ever had sexual intercourse. Male condoms were the most popular form of contraception purchased, used by 190 of the 274 (69%) participants who used contraception at last sexual intercourse. Of the participants indicating that they used a modern contraceptive at last sexual intercourse (n=263), 154 (59%, data not shown) had obtained it from a private, retail pharmacy (hereafter, 'pharmacy').

Of the 243 participants who were included in bivariate and multivariable analyses, 54% were male, 61% had attended secondary school or higher, and 70% were dependents (living with parents, grandparents or other older family members). A higher proportion of female participants than male participants were cohabiting, engaged, or married and had at least one child. Male participants had attended higher levels of schooling than female participants. Online supplementary table 1 presents selected characteristics of the 243 participants disaggregated by whether they obtained contraception at a pharmacy, shop or any other source: most shop users were male and purchased condoms.

### Who accesses contraception from pharmacies?
Bivariate analyses (table 3) indicated there was no evidence of an association between either age, sex or education and a young person's contraception being from a pharmacy. There was an association between pharmacy-purchased contraception and a participant's relationship status, and whether they had children. The greatest predictors of whether a young person had visited a pharmacy were the type of contraception they purchased and with whom they lived. Following multivariable analysis (table 3), there remained strong evidence of an association between pharmacy purchase of contraception and a young person's relationship status, living situation as well as the type of contraception they used. Young people living alone were almost twice as likely to have sourced contraception from a pharmacy as those living with a child or partner (adjusted prevalence ratio (PR) 1.96, 95% CI 1.07 to 3.59). Use of ECP remained the greatest predictor of a pharmacy purchase (adjusted PR 2.27 as compared with pill/injection use, 95% CI 1.21 to 4.27).

### Qualitative methods participant characteristics
Three FGDs were held with young men and three with young women—each FGD had approximately 10 participants. Of the 18 IDI participants, 10 were young women and 8 were young men. Female IDI participants had most recently purchased emergency contraception (n=7), injection (n=2) and condom (n=1). Male IDI participants had most recently purchased condom (n=6) and ECP (n=2).

Of the 19 key-informant participants, 10 interviewed pharmacy personnel were women and 9 were men. Participants were not probed in detail on their formal training (and therefore whether they should be operating in their current role). That said, we could ascertain that 13 of the participants had an appropriate amount of training for their reported tasks, and 4 did not (the final 2 were unclear). Self-reported education ranged from having some secondary education to full training as a pharmacist or pharmaceutical technologist. One participant was a nurse. Stakeholder demographics are not described to ensure they remain unidentifiable.

### Why are pharmacies appealing?
Participants indicated that it was a combination of the pharmacy *outlet*, the pharmacy *personnel* themselves and the *services* provided by the pharmacy which together made these establishments the preferred source of contraception for many young people (table 4).

Pharmacy outlets were appealing because of the convenience and anonymity they offered young clients. Pharmacies were located where young people lived, worked and spent time, making them easy contraception access points. If one pharmacy lacked what a young person was looking for, it was a short trip to the next one. 'Convenience' also extended to the days and hours pharmacies were open. This made them especially important on days

**Table 2** Baseline characteristics

| | All surveyed participants (n=740) | | |
|---|---|---|---|
| | **Female** | **Male** | **Total** |
| Ever had sexual intercourse | 231/347 | 281/393 | 512/740 (69%) |
| Used any contraception at last sexual intercourse | 126/231 (55%) | 148/281 (53%) | 274/512 (54%) |
| Used a modern contraceptive at last sexual intercourse | 118/231 (51%) | 145/281 (52%) | 263/512 (51%) |
| Used pharmacy-available contraception* | 116/231 (50%) | 143/281 (51%) | 259/512 (51%) |
| Where contraception was obtained | (n=116) | (n=143) | (n=259) |
| Pharmacy | 63% | 56% | 59% |
| Shop | 5% | 17% | 11% |
| Public dispensary or health centre | 13% | 7% | 10% |
| Hospital | 11% | 6% | 8% |
| NGO, private doctor | 3% | 4% | 4% |
| Community-based distributor, school, supermarket | 1% | 2% | 2% |
| Other person† | 1% | 4% | 3% |
| Other source (not specified)/don't know† | 3% | 3% | 3% |
| | Included participants using pharmacy-available contraception (n=243) | | |
| | Female (n=111) | Male (n=132) | Total (n=243) |
| Age | | | |
| 18–19 | 17% | 18% | 18% |
| 20–24 | 83% | 82% | 82% |
| Education (highest level attended) | | | |
| Primary or below | 54% | 27% | 40% |
| Secondary | 38% | 55% | 47% |
| Post-secondary | 8% | 18% | 14% |
| Relationship status | | | |
| Single | 23% | 42% | 33% |
| Friends with benefits | 3% | 8% | 5% |
| Dating | 42% | 42% | 42% |
| Cohabiting | 3% | 1% | 2% |
| Engaged | 9% | 5% | 7% |
| Married | 20% | 3% | 11% |
| Any children | | | |
| No | 74% | 92% | 84% |
| Yes | 26% | 8% | 16% |
| Living situation | | | |
| Lives alone | 8% | 23% | 16% |
| Lives with family (dependent) | 66% | 73% | 70% |
| Lives with child or partner | 26% | 4% | 14% |
| Contraception used‡ | | | |
| Male condom | 56% | 86% | 72% |
| Female condom | 4% | 2% | 2% |
| ECP | 20% | 6% | 12% |
| Daily contraceptive pills | 5% | 2% | 3% |
| Injection | 16% | 5% | 10% |

*These included male or female condom, emergency contraception (ECP), daily contraceptive pills and injectable contraception.
†These were excluded from analysis.
‡Participants could enter one contraceptive method.

**Table 3** Bivariate and multivariable analysis to identify personal characteristics that may be associated with a young person obtaining contraception from a pharmacy (vs any other source)

| | Purchased contraception from pharmacy | Unadjusted prevalence ratio (PR) (95% CI) | P value* | Adjusted PR (95% CI) | P value |
|---|---|---|---|---|---|
| All | 153/243 (63%) | | | | |
| **Age** | | | | | |
| 18–19 | 27/43 (63%) | Ref | 0.979 | | |
| 20–24 | 126/200 (63%) | 1.00 (0.78 to 1.29) | | | |
| **Sex** | | | | | |
| Male | 80/132 (61%) | Ref | 0.405 | | |
| Female | 73/111 (66%) | 1.09 (0.90 to 1.32) | | | |
| Education | | | | | |
| Primary or below | 60/96 (63%) | Ref | 0.904 | | |
| Secondary or above | 93/147 (63%) | 1.01 (0.83 to 1.23) | | | |
| **Relationship status** | | | | | |
| Single | 46/81 (57%) | 0.76 (0.61 to 0.94) | 0.0013 | 0.75 (0.61 to 0.93) | 0.0284 |
| Dating/'friends with benefits' | 86/115 (75%) | Ref | | Ref | |
| Married/engaged/cohabiting | 21/47 (45%) | 0.60 (0.43 to 0.84) | | 0.95 (0.67 to 1.35) | |
| **Children** | | | | | |
| No | 139/204 (68%) | 1.89 (1.24 to 2.92) | 0.003 | 1.25 (0.80 to 1.97) | 0.318 |
| Yes | 14/39 (36%) | Ref | | Ref | |
| **Living situation** | | | | | |
| Lives alone | 30/39 (77%) | 2.62 (1.51 to 4.53) | 0.0024 | 1.96 (1.07 to 3.59) | 0.0119 |
| Lives with family (dependent) | 113/170 (66%) | 2.26 (1.33 to 3.85) | | 1.53 (0.84 to 2.82) | |
| Lives with child or partner | 10/34 (29%) | Ref | | Ref | |
| **Contraception used** | | | | | |
| Condom (M/F) | 120/181 (66%) | 2.36 (1.34 to 4.14) | 0.0014 | 1.87 (1.02 to 3.43) | 0.0224 |
| ECP | 24/30 (80%) | 2.84 (1.59 to 5.09) | | 2.27 (1.21 to 4.27) | |
| Pills/injection | 9/32 (28%) | Ref | | Ref | |

*Any variable with p values <0.2 in bivariate analysis were included in the multivariable analysis.
ECP, emergency contraception.

where health facilities were known to be busy, or evening and weekend hours when young people might need contraception.

In addition, the relative privacy offered by pharmacies was especially important to young clients. Participants perceived pharmacies, with interactions limited to a pharmacy attendant and a client, to be far more discreet than similar services offered at public health facilities. Public health facilities had public waiting areas where young people may see someone they knew. In addition, services in the health facility might be categorised by service type (eg, contraceptive services separated from immunisation services, etc). This left young clients feeling particularly exposed should they need to walk up to a labelled 'family planning' window or step forward if a public announcement about contraceptive services was made.

The individuals behind the counter, and how they interacted with young people, were additional reasons young people preferred to obtain contraception from pharmacies. Pharmacy personnel were perceived to be established, fellow community members. Young clients appreciated seeing the same familiar faces, with less of the personnel turnover associated with public health facilities. When personnel were a similar age to young clients (a very strong preference of all young participants), many reported being able to communicate openly with pharmacy personnel and being more comfortable interacting with them.

Pharmacy personnel were perceived to be non-judgemental compared with those working in health facilities. There was a perception that a trip to a facility would result in difficult questions and a possible refusal to provide the desired contraceptive. Pharmacy personnel, by contrast, would treat young people 'well'. That is, they would provide the desired contraceptive without interrogation. Several participants speculated

**Table 4**  Reasons why pharmacies are appealing (selected excerpts from qualitative data)

| Outlet appeal | The physical pharmacy environment and its operation |
|---|---|
| Convenience (locations and hours) | "The chemist is near and whenever you want it [family planning] you can access it, anytime." Female pharmacy purchaser: injection |
| | "The good thing with chemist is that they are many of them… when you missed a certain contraceptive at a certain chemist you can go to the next chemist because they are several of them, not like the hospital." Female community member (FGD) |
| | "Yes, majority of them [young people] don't live near health centres. Second, health centres are usually busy. And it's not every day they [can be] attended to: there are specific days they have clinics… [The client] won't be able to make it there… even if the treatment was free. But there is a chemist—[they] can go for similar services." Pharmacist |
| Anonymity | "At the chemist there are not many people. I may go to Diani dispensary [a local public health facility], and there is someone who knows me and I go for family planning. I saw it would be better to go the chemist because I know that will be my secret and the attendant." Female pharmacy purchaser: emergency contraception |
| | "When you go to the facility, when you go to the FP room, everyone knows that you've gone to get FP. For young people [especially] because no one will want to see me—I'm 18, I'm 16 and I'm already using family planning. I'm not supposed to be sexually active. The kind of population that is in those FP areas, around those FP areas it's your mothers who are either breastfeeding, or they're pregnant and have gone for ANC." Ministry of Health official, County level |
| **Personnel appeal** | **The person behind the counter** |
| Interpersonal relationship | "The chemist is just within the neighborhood and I know the guy he is my friend outside job so it wasn't stressful for me in fact it was really fast and easy." Male pharmacy purchaser: ECP and condoms |
| | "The person in charge is my friend, I can go to him with my problems and he would assist me, he is not that far for me to reach him with my phone—he is my neighbor I could have a problem even at night and be able to reach out to him." Male pharmacy purchaser: ECP |
| Seen as part of the community | "I chose it because it has been there for many years even before I was born till the time I finished school. The attendants are just normal. Many people get help from there so I saw it good to also go there." Female pharmacy purchaser: ECP and injection |
| | "What I had said about the hospital, when you get there you will find the person who served you before is transferred but when you come to the chemist you will find the person that served you before." Female community member (FGD) |
| Non-judgemental | "I thought at the chemist they will understand me and I would talk to them [better] than at the hospital where they will say I do not need to use those things or even talk to me harshly." Male pharmacy purchaser: ECP and condoms |
| | "At the chemist, that person wants—since it is a business—[to] just give, as compared to the hospital where when you get there you will find nurses who are arrogant or other doctors who will insult you." Male community member (FGD) |
| **Service appeal** | **The contraception-purchasing transaction** |
| Speed | "You know at the dispensary it is a must you meet with the doctor for more explanation. And maybe there is a service you need to pay for, the expenses are many at the dispensary unlike the chemist where everything is fast, when you get there you get what you want and leave." Youth female, has purchased ECP and condoms |
| | "You get in a hospital, there are so many people queueing outside that are waiting to see a doctor. Here comes a young lady who is in a hurry. That particular person will find it more convenient to go to a chemist shop rather than going to a hospital." Pharmacist |
| Cost | "It is not easy for the government hospital. It is best, if you have money, you go to private hospitals. Now that is why you see if someone does not have money, or us the young people, we just go to the chemist because there is no cash to see a doctor for Ksh 600. At the chemist you just go direct and you are served." Male pharmacy purchaser: ECP and condoms |
| | "Chemists are not expensive like hospitals. In hospital you can be told it is a government hospital but you end up being asked to give out a lot of money. In [the] chemist the money you get asked is for [paying for] P2 [an emergency contraceptive], yah but in hospital you will be told to do some test because we think it is this and this." Female pharmacy purchaser: ECP |
| | "Free does not always mean free. Sometimes, something will be free, but by the time you get it, the process is a lot. Because for us, we don't just offer family planning, we do [mandatory] counselling. The person who is going to a chemist is someone who has made up his or her mind. But in the public facilities, you are counselled, you are explained to, you are told the different methods, then you are given a chance to make an informed choice. So, I think that… is a barrier somehow." Ministry of Health official, County level |

that the for-profit aspect of pharmacies could be a reason that they were treated better and not refused services.

Finally, pharmacy contraception services themselves were appreciated for being fast and cheap. Participants routinely referenced the queueing for services and long wait times driving young people away from health facilities and into pharmacies instead. Services were also perceived to be cheaper than both private health facility services as well as public health facility services. Private health facilities were considered out of financial reach for most young people—making a pharmacy a more affordable option. However, at public health facilities, where contraception-related services are meant to be free, participants indicated that this was often not the case in practice. Expenses related to travel or 'tests' (eg, a pregnancy test) ordered by healthcare providers prior to dispensing contraception made real costs related to public services difficult to predict. Finally, as one government official acknowledged, even when services were free, the time and processes required could deter young people who knew what they wanted from going to facilities.

## DISCUSSION

This mixed-method study determined pharmacies to be the most popular source of contraception for young people in a peri-urban area of Kwale County. In total, 59% of participants (and 63% female participants) who had ever had sex and self-reported use of a modern contraceptive at last sexual intercourse had obtained their contraception from a pharmacy. This is higher than previously reported for Kenya as a whole.[8] Multivariable analyses indicated that young people who were living alone relied more heavily on pharmacies for contraception more than their peers. That said, the strongest predictor of young people's contraception coming from pharmacies was the type of contraception they used, specifically emergency contraception. Qualitative findings demonstrated that young people valued pharmacies for their convenience, anonymity, non-judgemental and personable staff, service speed, as well as predictable and affordable prices.

Together, these mixed methods indicate that pharmacies provide a valued source of contraception for those young people who may face increased scrutiny or gatekeeping in health facilities. For young people using condoms or ECP, the reported convenience and speed of service explains the strong preference for pharmacies. Following unprotected sex, a young person needing ECP would understandably prefer to pay for it at a nearby pharmacy instead of travelling to a healthcare facility, waiting in line and negotiating with a possibly reluctant health worker to obtain it for free (assuming the public facility stocked ECP[20]).

This study had several limitations. In the survey, participants were asked to specify where they or their partner had obtained the contraception used at last sexual intercourse. This question is standard in studies looking to establish contraception prevalence. However, our not further ascertaining whether it was the respondent or their partner *who picked up the contraception* affected our ability to distinguish differences in preferred sources between young men who obtain contraception versus young women who obtain contraception. Second, to recruit young people who had recently purchased contraception from pharmacies, we relied on assistance from five pharmacies, purposively selected. It is possible that young purchasers patronising different pharmacies might have had different experiences than those captured here. Finally, our youth participants in focus group discussions may have felt uncomfortable discussing contraceptive use in a group; we attempted to mediate this by structuring discussion around vignettes of 'typical' young people. This study is strengthened by its mixed-methods design and its use of multiple qualitative methods, and inclusion of both pharmacy personnel and young people to triangulate research findings on a sensitive subject.

Our quantitative findings differ substantially from an analysis of Kenya's DHS (KDHS) data, which found that nationwide, 13% of Kenyan women aged 15–24 currently using contraception reported obtaining it at a commercial drug seller.[8] There may be several reasons for this, in addition to the four years between the KDHS and our own data collection. Our study area was a peri-urban setting while the DHS analysis uses nationwide data. Over 70% of Kenya's population is rural.[21] Finally, our study's inclusion of ECP and measuring contraception use at last sexual intercourse (rather than 'current use') is also a likely contributor. Twelve per cent of participants in this study used ECP at last sexual intercourse, and the KDHS did not specifically capture ECP use.[22] The DHS's measures of contraception 'current use' in general has been previously critiqued for not being able to capture contraceptive methods which may be used periodically, including ECP.[23] Our link between ECP purchasers and pharmacies is in line with earlier data from urban Kenya, which indicated that upwards of 96% of adult women needing ECP obtained it at a pharmacy.[24]

By contrast, our qualitative findings were largely in line with previous research. One systematic review featuring studies mostly from high-income countries (HICs) affirms that young people appreciate pharmacies for their convenience, speed of service and ease of contraception access.[9] However, this review also reported mixed evidence (all from HICs) as to whether pharmacy services were considered 'private',[9] while our study found an almost universal appreciation of pharmacies for their anonymity/privacy. This difference may be a result of different dispensing protocols and establishment layouts in pharmacies and public health facilities in HICs versus LMICs. Evidence from other LMICs corroborates our findings that among young people,[25] and the general population,[26] pharmacies' contraception services are appreciated for the privacy offered.

While this study focused on pharmacies, its findings also cover perceptions around how contraception services are delivered to young people in public health facilities. Pharmacies were naturally contrasted with health facilities when participants explained young people's preferences and were perceived to be everything that health facilities were not: fast, private and non-limiting. The extra 'procedures' required to obtain contraception in health facilities—which in many cases are unnecessary[27] and have been demonstrated in other settings to limit access[28 29]—were especially unwelcome for young people, who were uninterested in extended counselling and wary of laboratory tests. As a result, pharmacy services were deemed more 'predictable' than those obtained in health facilities (public or private).

For Kenya, pharmacies are likely to remain a preferred choice of contraception as long as barrier methods and short-acting forms of contraception are popular with young people.[22] Policy-makers should therefore recognise their role as contraception providers, especially for a community's younger members. Finding ways to link the myriad licensed pharmacies to focal points in public health facilities could strengthen a supportive 'network' of accessible and appealing contraception services available to young people. A similar hub-and-spoke approach is used in the implementation of Kenya's

broader Community Health Strategy, where community health volunteers are embedded within the community and report back to a facility-based community health extension worker.[30] Such a system, complemented by improved adolescent-friendliness of public health facilities, would also enable easier referral of young people to providers who can offer them more effective forms of contraception. However, none of this can succeed without taking needed steps to improve pharmacy regulation, personnel training and the overall quality of services.[31]

Our data revealed that shops were the second most popular source of contraception for young men. The reliance on shops and lower-level drug dispensaries is seen elsewhere in the region: one survey in Nigeria found that among young people aged 15 to 24, around half sourced their contraception from 'chemists/patent medicine shops' (a cadre of establishment below pharmacies, which does not exist in Kenya).[32] Unfortunately, exploring shops in further detail was beyond the scope of our data collection. Additional research is needed to understand how to incorporate these more informal sources into contraception interventions. That said, integrating them into the broader 'network' of contraception providers for young people will be even more challenging: lower-level drug dispensers are only peripherally associated with the health system in many settings, while shops are not associated at all.

Finally, we must acknowledge those still left behind. Of participants who reported ever having sex, almost half of them (49%) had *not* used any modern contraception at last sexual intercourse. Aside from those who wish to conceive, these are young people who are not being reached by the current network of public and private health facilities, pharmacies and even neighbourhood shops. They are a reminder that improving the quality of services in these outlets is necessary but not sufficient to address young people's contraceptive needs. There is a continued need for multi-sectoral interventions, including comprehensive sexuality education, to increase demand for contraception among youth (dispelling myths, addressing taboos and stigma, and increasing agency),[33] address barriers to accessing it (including community norms around acceptability)[3] and promote uptake of highly effective forms of contraception.

Young people in Coastal Kenya steadily rely on pharmacies for contraception and often prefer them to health facility services. Many of the pharmacy qualities most appreciated by young participants are also hallmarks of youth-friendly health services, which should be available in any outlet a young person attends for health services.[19 34] If a young person chooses to use modern contraception, their selection of an outlet will be determined by several factors, including the type of contraception desired, living situation and relationship status. Collaboration between health facilities and retail pharmacies at local levels can exchange operational strengths between these providers. Then, wherever a young person presents for contraceptive services, they encounter one part of a supportive network of quality providers.

**Author affiliations**
[1]Department of Sexual and Reproductive Health and Research including UNDP/UNFPA/UNICEF/WHO/World Bank Special Programme of Research, Development and Research Training in Human Reproduction (HRP), World Health Organization, Geneva, Switzerland
[2]Swiss Centre for International Health, Swiss Tropical and Public Health Institute, Basel, Switzerland
[3]University of Basel, Basel, Switzerland
[4]International Centre for Reproductive Health Kenya, Mombasa, Kenya
[5]Department of Human Anatomy, University of Nairobi, Nairobi, Kenya
[6]Department of Public Health and Primary Care, Ghent University, Ghent, Belgium

**Acknowledgements** We appreciate the support of Jefferson Mwaisaka and Winnie Wangari. Our sincere thanks to all data collectors and participants.

**Contributors** LG conceived of the study and developed the protocol with substantive input from KW and AMH. PG was Principal Investigator of the ARMADILLO study and thereby supported LG in setting up this study's infrastructure in Kenya. LG trained and supervised data collectors, with guidance from PG. JAC and MW developed the statistical analysis plan. LG led the manuscript writing with substantive input from KW and AMH. All authors reviewed and edited drafts.

**Funding** This work was partially supported by the UNDP/UNFPA/UNICEF/WHO/World Bank Special Programme of Research, Development and Research Training in Human Reproduction (HRP).

**Disclaimer** LG and JAC are staff members of the World Health Organization. The named authors alone are responsible for the views expressed in this publication and they do not necessarily represent the views, decisions or policies of the World Health Organization.

**Competing interests** None declared.

**Patient consent for publication** Not required.

**Ethics approval** This study received ethics approval from the Ethikkommission Nordwest- und Zentralschweiz (EKNZ) (Req-2017-00389) in Basel, Switzerland, as well as the University of Nairobi/Kenyatta National Hospital in Nairobi, Kenya (P274/05/2017). The ARMADILLO RCT also received ethics approval from WHO (Protocol WHO A65892) and is registered with the ISRCTN Registry (ISRCTN85156148).

**Provenance and peer review** Not commissioned; externally peer reviewed.

**Data availability statement** Data are available on reasonable request. The full deidentified quantitative dataset can be made available on request to the corresponding author. Qualitative data cannot be shared publicly, as consent procedures for participants did not include making full interview and focus group discussion transcripts publicly available. However, transcript excerpts are available to researchers on request to the corresponding author and following approval from the University of Nairobi/Kenyatta National Hospital Ethics Committee (contact via uonknh_erc@uonbi.ac.ke).

**ORCID iDs**
Lianne Gonsalves http://orcid.org/0000-0003-2409-5043
Jenny A Cresswell http://orcid.org/0000-0002-9553-1132
Michael Waithaka http://orcid.org/0000-0002-6766-9492
Peter Gichangi http://orcid.org/0000-0001-9636-165X

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
