## [Reviewer comments · BMJ Open]

ARTICLE DETAILS

TITLE (PROVISIONAL)	Mixed-methods study on pharmacies as contraception providers to Kenyan young people: who uses them and why?
AUTHORS	Gonsalves, Lianne; Wyss, Kaspar; Cresswell, Jenny; Waithaka, Michael; Gichangi, Peter; Martin Hilber, Adriane

VERSION 1 – REVIEW

REVIEWER	Efi Mantzourani Cardiff University, UK
REVIEW RETURNED	21-Nov-2019

GENERAL COMMENTS	Thank you for the opportunity to review this study. It presents an interesting topic. I would recommend substantial redrafting before the manuscript is ready to be considered for publication, to improve transparency of methodology, and to ensure the tone of discussion is appropriate and does not overestimate the results. Some specific comments: Abstract Background – please revise passive language to improve readability (also in l19 of introduction). Please include specific background to study e.g. situation in Kenya. L38 – suggest changing “struggle” (implying services are active) to “fail” Design and setting – please remove all results from this section and revise accordingly. Please add instead actual methods of analysis of your different types of data. Participants: how were they identified? Sampling frame? Results – please revise with information moved from design. Not sure what “contraception” in l11 refers to: is it barrier contraception, routine oral contraception, something else? Suggest changing word “sex” to “sexual intercourse” throughout the manuscript. Conclusions – not clear if authors refer to prior reported use in Kenya, or from literature internationally, this should be taken into account as context within which to explain the results “The phrasing of survey questions affected our ability to distinguish differences between young men versus young women who obtain contraception” – not sure what that means, as data on sex was collected based on Table 2. Becomes clearer in discussion, but text here needs to be revised. Introduction L5 – please define age range you are referring to Methods
--

	Please see comment in abstract, actual number of participants, focus groups etc is not methodology, it is result. Please revise the whole section and table accordingly. It would be helpful to have the specific section of the survey as an appendix. Surveys: What was the study population? Who identified potential participants? How was the survey distributed, what options were available for participation (e.g. electronic?), was the survey piloted, what were the constructs, did it involve open/closed questions? What were the inclusion criteria mentioned in table on page 17? What is an “enumerated” household, and is that selection representative of entire population? Qualitative: What was the sampling frame for participant recruitment in focus groups and interviews? How were potential participants approached for key informants, by whom? What was the topic guide (please include in an appendix). Was it piloted? PPI – the whole section needs to be written in narrative format Analysis of quantitative data – who entered data, any QA? How did you deal with missing data? Analysis of qualitative data – any back-translation for quality assurance? How did you deal with anonymisation of the data? What was the quality approach and research paradigm? How about researcher characteristics and reflexivity? I suggest that the authors reflect on the standards for reporting qualitative research to improve transparency of the methods/results. Results Quantitative – how many people were approached, of whom 740 responded? Essentially, what is the response rate? Information presented in table on page 17 (no legend for the table) but not here. Qualitative – Table of participant characteristics needs to be presented here. Not clear which of these themes are down to deductive and which down to inductive analysis. A narrative at the start of each theme would be very helpful for the reader, rather than a long text, to indicate also numbers of participants who made statements that supported each theme. Discussion I think the main discussion point of the authors needs to be toned down, as it is not directly (or obviously) justified by the results. Not clear if selection of households is representative. How many people use services in pharmacies in relation to population overall or other settings? From lines 12-17 it appears that there is a big % of population with unmet needs, so the authors are effectively getting opinions of population with needs that are met by pharmacies – is that representative? Lines 16-17 of discussion – not sure the findings differ substantially from this, due to argument above. Lines 48-49 – where else can young people get EC in Kenya? This type of contraception is particularly associated with stigma in literature when other healthcare settings are visited, as often the services are not anonymous. Some discussion around this needs to be added. Flow of text needs to be improved, with consistent arguments that run through. Please ensure language throughout is formal, e.g. line 48 “all told”.
--	---

REVIEWER	Andy Husband Newcastle University England
REVIEW RETURNED	29-Nov-2019

GENERAL COMMENTS	Thanks for the opportunity to review this paper, please consider the following comments which are a mix of pedantry and substantive issues. Abstract The term 'at last sex' I'm not sure is accurate, I would suggest the term 'sexual intercourse' may be better in this sentence. Equally the paper seems to focus entirely on contraception for birth control as opposed to protection against sexually transmitted disease, which I accept is only relevant for certain barrier methods. Might it be worth framing the study a little more clearly around birth control and heterosexual sexual intercourse? Im presuming that is the focus from what I have read, albeit it is not explicit. Introduction Line 6 - I would argue this needs to be referenced, how do we know contraceptive services are not meeting the need? Im not sure that is entirely true, it may be that pharmacies are more accessible and therefor convenient, which is fine. I would like to see a reference source for this statement. Line 12 - might be worth a word about the age of consent for sexual intercourse in general in many of the countries you are discussing here. In the UK that age is 16 years and the 15-19 age bracket obviously includes part of that period. I would argue this may help frame the introduction from that viewpoint. Methods Starting at Line 40, I would argue this section should be in the introduction, I accept it is framing the sample but one might also argue it is addressing why you have chosen to undertake this study in this particular setting. Line 48 - define SRH on first use Table 1 - I would argue this is results, the demographics of the sample are a key aspect of what you found and how your sample was composed. My general feeling is that the methods need to be re-written. I accept it is a nested part of a larger study but that should not prevent an accurate précis of what was done in this study. I am interested in the fact this is a mixed-methods study yet does not appear to demonstrate a link between the quant and qual aspects. How did the survey inform the focus of the qual (as would be the case in true mixed methods) or was this a case of a parallel quant and qual study ? P5 Line 4- how was a random set of pharmacies generated? Did you use a specific approach to randomisation ? Once this is clarified there is also a need to clarify the location of the pharmacies, were the pharmacies chosen from a variety of
---

	neighbourhoods and therefore representative of different sociodemographic groups ? What theoretical approach was taken to the interviews? , What analytical framework was used? I can see a reference to data saturation in the text. There is very little in terms of how you have gone about triangulation of the qualitative data to manage bias? Results You use the term 'modern contraceptive' for the first time here. Im presuming thesis to differentiate between other methods of contraception. Might it be worth defining what you mean here clearly in the introduction ? Characteristics - I find this difficult as I don't know which contraceptive products are available from non-medical outlets (including pharmacies) in Kenya. You have not framed that in the introduction. Are all of these products widely available from other sources or are they limited to pharmacies. The relationships are therefore more obvious if there are only medical outlets to use to obtain contraception. In terms of the presentation of the qual data Im not clear what themes were identified in the thematic analysis. If the themes in the table are those I would argue some of these are essentially the same thing. What is the difference between service appeal and convenience ? Are these things not generally about access? I would argue the paper needs a conclusion
--	--

REVIEWER	Paula Tavrow Fielding School of Public Health, UCLA USA
REVIEW RETURNED	05-Dec-2019

GENERAL COMMENTS	Comments on: “Mixed-methods study on pharmacies as contraception providers to Kenyan young people: who uses them and why?” by Paula Tavrow This article makes an important contribution to the literature on adolescent contraceptive use in Africa—specifically where Kenya young people obtain their contraceptive methods and why so many rely on pharmacies, even where health facilities are free. I enjoyed reading it, particularly the comments from the young people, which were well-selected and organized. Having been working in Kenya for the past two decades, I also appreciate that this topic needs more attention. The main weaknesses of the paper center on its imprecision, inattention to the implications of this reliance on pharmacies, and general readability. In my review, I will seek to highlight the areas of imprecision and pose some questions to be answered. Regarding readability, suffice to say that the authors frequently use compound sentences with colons or semi-colons, when simple
--

declarative sentences would serve them better. The paper should be re-written to remove colons and semi-colons except where strictly necessary.

Also, the discussion is rambling and overlong. It should be broken into discreet topic areas with sub-headings. Limitations should appear near the end, followed by conclusions.

Concerning imprecision, the authors frequently make statements that are not borne out by the paper's data or do not convey the issue accurately. Here are two prime examples from the abstract:

- **“Public sector contraceptive services often struggle to meet the needs of young people around the world. Instead, private pharmacies have been demonstrated to be a relied-upon source of modern contraception for young people.”** This is misleading. Public sector services are not “struggling” to meet the needs of young people. Instead, they are often very unfriendly to young people, as well as inconvenient, not confidential and not private. These are the main reasons that young people visit pharmacies instead. Pharmacies have not “been demonstrated” to be sources of contraception. They become the sources because young people find them more acceptable.
- **“Among surveyed participants, 59% had used contraception purchased from a pharmacy at last sex.”** This is inaccurate. It should read: “Among surveyed participants who had used a modern form of contraception at last sex, 59% reported that they had purchased the method from a pharmacy.” The distinction is important. If one starts with “all surveyed participants (N=740),” one must conclude that 21% (153/740) reported that they had used contraception purchased from a pharmacy at last sex (based on Table 2). Also, the authors need to note that this is *as reported* by the young people. Those living at home may be reluctant to admit to ever having sex or ever using contraception.

Note that this “59%” figure also is the starting point of the Discussion. But, again, this is misleading. Again per the data in Table 2, *nearly half* of young people who are having sex are not using any contraception at all, mainly because of psycho-social barriers (including fears of side effects, fears of others knowing they are sexually active, etc.). Of those who do use modern contraception, 59% are using methods purchased at pharmacies, which are the *least effective in preventing pregnancy in actual use* (condoms and EC pills). More on this issue later.

Another way that the paper is imprecise is in Table 2. For baseline characteristics, it would be better to show the results by gender, particularly if the desire is to compare the results to the Kenya DHS, which presents its data by gender. Even if there is a limitation in the data that it is not known who supplied the contraceptive method (for condoms), it still would be more useful to view the data by gender.

Showing the data by gender would increase precision for the subsequent discussion. Also, I do not understand why the age data was bifurcated into “18-19” and “20-24”. I think that it would

be best presented in thirds (e.g., 18-20, 21-22, 23-24) to see if rising age changes some of the contraceptive use dynamics.

It was curious to have a supplementary table that is not in the main text, when it was addressing one of the key study questions. I believe it would be best to have data presented as “retail (pharmacy/shop)” vs “non-retail (health centres, etc.), and to have the rows sum to 100% (not the columns). This would help us to see who uses retail vs. non-retail, as per the study question. In my view, separating out shops from pharmacies isn’t that useful, because some young people cannot tell the difference. The real question is: who uses retail rather than public/NGO clinics?

Another area of imprecision surrounds the “relationship status.” How is “single” different from “dating”? Since both are sexually active, does “single” signify that the last sex was a “one night stand” or “transactional sex”? This would be useful to differentiate. Are those in a “dating” relationship more likely to use clinics and hormonal methods?

Regarding implications, this is where the authors miss the boat in my opinion. Why are young people turning to pharmacies for their supplies? It is mainly owing to three reasons: health facilities are not youth-friendly, community norms are not accepting of sexually-active unmarried people, and comprehensive sexuality education in schools is generally non-existent. Hence, young, unmarried, nulli-parous people seek the convenience and “privacy” (by which they mean the anonymity or not being seen by the community) of pharmacies to get those modern methods which they believe are least likely to be “harmful”. Without good information about implants, injectables, IUCDs, and oral contraceptives, the young people fall back on condoms and ECPs. Yet both have only 80-85% efficacy in real use. This leads to very high rates of unwanted pregnancies and illicit abortions—both of which surprisingly are unmentioned in the paper.

For their discussion, the authors need to look not only at those who visit pharmacies, but also at those who do not, to get a clearer picture of what is happening. Their discussion needs to discuss this phenomenon. Namely, married young people, and/or those with children, are more likely to go to clinics because it is socially acceptable to do so. They are thus able to get more effective contraception. On the other hand, nurses are averse to providing ECP to girls more than once. They feel it should only be for “emergency use” (even though it is actually fine to use frequently, just less effective). For girls who rely on ECP, they therefore need to get it from pharmacies.

Lastly, the above realities should affect the conclusions of the authors. The pharmacy “solution” is not optimal for the reasons I laid out. Yes, it would be better if pharmacies could offer a more comprehensive range of methods. But the health services still do need to be improved and comprehensive sexuality education (that includes contraceptive methods) needs to be given in the schools, so that young people are aware of all of their options and do not need to rely mainly on pharmacies, which provide limited options for family planning. The current system is not only less affordable for young people (and may create greater health disparities since low income will have reduced access) but is also more prone to contraceptive failure since it relies on condoms and ECPs. The

	ensuing social consequences--which may include unwanted births, dropping out of university, and botched abortions--can have dire long-term effects, particularly for girls.
--	---

VERSION 1 – AUTHOR RESPONSE

Reviewer(s)' Comments to Author:

Reviewer: 1

Reviewer Name: Efi Mantzourani

Institution and Country: Cardiff University, UK Please state any competing interests or state 'None declared': None declared

Please leave your comments for the authors below Thank you for the opportunity to review this study. It presents an interesting topic. I would recommend substantial redrafting before the manuscript is ready to be considered for publication, to improve transparency of methodology, and to ensure the tone of discussion is appropriate and does not overestimate the results.

Please see our answers below on the specific comments around transparency and methodology.

Some specific comments:

Abstract

Background – please revise passive language to improve readability (also in l19 of introduction).

Please include specific background to study e.g. situation in Kenya.

We acknowledge this comment however, per abstract requirements, have been instructed to delete the background section entirely.

L38 – suggest changing “struggle” (implying services are active) to “fail”

Comment addressed and wording modified.

Design and setting – please remove all results from this section and revise accordingly. Please add instead actual methods of analysis of your different types of data.

Thanks for pointing to this. We would however like to indicate that there are no results in the methods section. It is standard practice, when presenting qualitative results, to put the number of data collection events in the methods section. Consequently, we have not modified and hope this is acceptable to the reviewer

Participants: how were they identified? Sampling frame?

Thanks for this observation. Based on the reviewer's comments, we have further elaborated the sampling in the Methods section of the manuscript. Given the word limits in the abstract we have however not further elaborated in the abstract the sampling.

Results – please revise with information moved from design. Not sure what “contraception” in l11 refers to: is it barrier contraception, routine oral contraception, something else? Suggest changing word “sex” to “sexual intercourse” throughout the manuscript.

The comments of the reviewer on terminology are appreciated. Consequently, contraception has been revised to say 'modern contraception' and sex and has been revised to 'sexual intercourse' in the abstract. In the body text we have expanded on what 'sexual intercourse' is implied to mean to participants on Line 180 (as it was never explicitly defined – not uncommon for these kinds of questions – for participants). 'At last sexual intercourse' is later abbreviated to 'at last sex' (a common phrase used in sexual and reproductive health literature) on Line 210.

Conclusions – not clear if authors refer to prior reported use in Kenya, or from literature internationally, this should be taken into account as context within which to explain the results “The phrasing of survey questions affected our ability to distinguish differences between young men versus

young women who obtain contraception” – not sure what that means, as data on sex was collected based on Table 2. Becomes clearer in discussion, but text here needs to be revised.

Thanks for this useful note. We have clarified in the conclusions section that we are talking only about Coastal Kenya. We have also revised the ‘young men vs young women’ comment in the strengths and limitations section.

Introduction

L5 – please define age range you are referring to

Comment well taken and we provide now indications that the study refers to the age range 18-24.

Methods

Please see comment in abstract, actual number of participants, focus groups etc is not methodology, it is result. Please revise the whole section and table accordingly.

As indicated above, the concern of the reviewer is well taken. We would however like to indicate that it is standard practice in qualitative research to present the number of data collection events (interviews, FGDs, etc – as well as the numbers of participants who were involved) in the methods section rather than the results section. This is confirmed by the SRQR checklist as well. This is not the norm for quantitative studies. That said, and factoring in the changes proposed by this reviewer to the results section (see first comment in Results section below), we feel the most straightforward and parsimonious way to present the data is to keep the table where it currently is. As a result we would like to keep the place on the number of participants as it stands and have not introduced specific changes

It would be helpful to have the specific section of the survey as an appendix.

Thanks for this suggestion – we are submitting relevant parts of the survey as well as qualitative question guides as part of the resubmit.

Surveys: What was the study population? Who identified potential participants? How was the survey distributed, what options were available for participation (e.g. electronic?), was the survey piloted, what were the constructs, did it involve open/closed questions? What were the inclusion criteria mentioned in table on page 17? What is an “enumerated” household, and is that selection representative of entire population?

This is a useful comment enabling us to strengthen the manuscript. We have modified the text in two areas of the Methods section, starting on Line 113 and then again on Line 142 to clarify appropriately. The surveys were not formally piloted but were tested for comprehension among 21 quantitative data collectors. We have included the relevant sections of the survey instruments as an Appendix so that readers of BMJ Open can access the information.

Qualitative: What was the sampling frame for participant recruitment in focus groups and interviews? How were potential participants approached for key informants, by whom? What was the topic guide (please include in an appendix). Was it piloted?

We acknowledge this comment of the reviewer and we clarify now in the paragraph (starting Line 123) that FGD and youth in-depth interview participants were purposively recruited. Purposive sampling is common for qualitative studies, as it helps to ensure that the smaller participant pool has the key attributes being studied (in the case of youth in-depth interviews, for example, having directly purchased contraception from a pharmacy themselves). The process of recruiting and approaching key informants is described in further detail starting from Line 130. We are including topic guides for all qualitative methods (FGDs, youth IDIs, and key informant interviews). Topic guides were not ‘piloted’ in the quantitative sense – instead during training of qualitative data collectors, each question was reviewed and discussed in detail for comprehension – based on data collectors’ feedback, wording was modified if needed.

PPI – the whole section needs to be written in narrative format

This is a helpful observation. We have revised our statement to make it narrative.

Analysis of quantitative data – who entered data, any QA? How did you deal with missing data?

The concern of the reviewer is well taken. We indicate, starting on Line 143, that data collectors entered data (on tablets) except for one small set of questions on sexual behaviour

where youth participants could enter the data themselves. For the small set of questions around which this analysis was based, there was no missing data.

Analysis of qualitative data – any back-translation for quality assurance? How did you deal with anonymisation of the data? What was the quality approach and research paradigm? How about researcher characteristics and reflexivity?

Thank you for this. There was no back translation of data. Instead, for a subset of interviews, an independent member of the research team (someone who did not conduct the interview nor transcribe/translate it) compared the English language transcript to the original Swahili language audio file. We have added this clarification in Line 192.

As this was mixed methods with the quantitative part playing a significant role in the study, we did not intentionally identify a qualitative approach and research paradigm – however, what we developed (qualitative methods alone) was a case study in a constructivist paradigm. Given BMJ Open's diverse readership, however, we have opted not to include these terms in the paper, as many readers may not be familiar with them. Instead, based on this comment and one from Reviewer 2, we have included information on the theoretical approach guiding qualitative analysis, which was to identify to what extent pharmacy services fill the five WHO-defined dimensions of 'quality health services' for adolescents.

We have also added a reflexivity statement (subheading 'Research characteristics and reflexivity')

I suggest that the authors reflect on the standards for reporting qualitative research to improve transparency of the methods/results.

The concern of the reviewer well noted. See response to the previous comment. Further, we have completed the SRQR checklist along the resubmission.

Results

Quantitative – how many people were approached, of whom 740 responded? Essentially, what is the response rate? Information presented in table on page 17 (no legend for the table) but not here.

This comment is appreciated, and per the suggestion, we have added information about the response rate to the very beginning of the results section. The information is also captured in the flow diagram of study participants (Figure 1)

Qualitative – Table of participant characteristics needs to be presented here. Not clear which of these themes are down to deductive and which down to inductive analysis. A narrative at the start of each theme would be very helpful for the reader, rather than a long text, to indicate also numbers of participants who made statements that supported each theme.

The concern of the reviewer is noted. We have added additional information qualitatively describing the participants (including sex, which was probably the key demographic missing), which we believe will help to contextualize the responses. This is available under Results subheading 'Qualitative methods participant characteristics'. We intentionally did not collect detailed demographic information from our key stakeholder/pharmacist participants, as we did not want to dissuade those who were not legally meant to be dispensing (unqualified attendants with no higher education, but also inappropriately-qualified professionals like nurses) from participating and being open with our data collectors. The same is true for recent purchaser youth (our IDI participants): while we assured them of the confidentiality of their participation, there was no demographic information (already having age range 18-24, and sex) that the research team thought critical enough to qualitative analysis that outweighed the importance of making our young purchasers feel comfortable.

We appreciate the request for clarification of themes identified in inductive vs deductive coding and have added notes clarifying this to the final paragraph of the Methods section (starting Line 197)

Discussion

I think the main discussion point of the authors needs to be toned down, as it is not directly (or obviously) justified by the results. Not clear if selection of households is representative. How many people use services in pharmacies in relation to population overall or other settings? From lines 12-17

it appears that there is a big % of population with unmet needs, so the authors are effectively getting opinions of population with needs that are met by pharmacies – is that representative?

'Unmet need for contraception' is a standard and widely-reported indicator in the contraception and broader sexual and reproductive health literature and is often used to demonstrate the state of 'SRH need' for an area or a population. It was presented in the Introduction only in this context. Given that BMJ Open has a broad readership, we have deleted this paragraph entirely to avoid similar confusion from readers. The entire Discussion has been substantially revised, so we hope that the reviewer's concerns are addressed under the new version.

Lines 16-17 of discussion – not sure the findings differ substantially from this, due to argument above. *Based on all reviewers' comments, we have substantially revised the discussion section. This said, we do not claim that our findings are representative of all young people. Particularly in the discussion, we have taken care to note where our findings are in line with other findings from the literature and where they differed (Discussion Paragraph 4 and 5).*

Lines 48-49 – where else can young people get EC in Kenya? This type of contraception is particularly associated with stigma in literature when other healthcare settings are visited, as often the services are not anonymous. Some discussion around this needs to be added.

We think that this is a very interesting point. Per this comment and another reviewer's we have added a reference to other sources of contraception in the Introduction (last sentence of paragraph 3). We also think that 'anonymity' captures the appreciated quality more specifically than 'privacy' and so have incorporated this in the results and discussion.

Flow of text needs to be improved, with consistent arguments that run through.

This concern of the reviewer is well noted. We have altered and improved the arguments along the manuscript substantially, particularly in the restructure of the Discussion, and hope by this to adequately address the reviewer's comment.

Please ensure language throughout is formal, e.g. line 48 "all told".

Reviewer's comment acknowledged and manuscript revised accordingly.

Reviewer: 2

Reviewer Name: Andy Husband

Institution and Country:

Newcastle University

England

Please state any competing interests or state 'None declared': None declared

Please leave your comments for the authors below Thanks for the opportunity to review this paper, please consider the following comments which are a mix of pedantry and substantive issues.

Many thanks for these comments and the review.

Abstract

The term 'at last sex' I'm not sure is accurate, I would suggest the term 'sexual intercourse' may be better in this sentence. Equally the paper seems to focus entirely on contraception for birth control as opposed to protection against sexually transmitted disease, which I accept is only relevant for certain barrier methods. Might it be worth framing the study a little more clearly around birth control and heterosexual sexual intercourse? I'm presuming that is the focus from what I have read, albeit it is not explicit.

We note the reviewer's comment and we've made the change to 'at last sexual intercourse' in the abstract. In the text, what 'sexual intercourse' is implied to mean to participants (as it was never explicitly defined in the survey– not uncommon for these kinds of questions) is described on Line 180 and hopefully more in line to addressing this reviewer's concern. 'At last sexual intercourse' is later abbreviated to 'at last sex' (a common phrase used in sexual and reproductive health literature) on Line 210.

Introduction

Line 6 - I would argue this needs to be referenced, how do we know contraceptive services are not meeting the need? I'm not sure that is entirely true, it may be that pharmacies are more accessible and therefore convenient, which is fine. I would like to see a reference source for this statement.

Thanks for this comment. We have moved up a reference to the continued unmet need for contraception among young women – an indicator often used to demonstrate sexual and reproductive health service (including contraceptive service) progress or lack thereof - to substantiate the comment. Additionally, the subsequent sentence highlights the well-documented reasons why young people struggle to get services from public facilities.

Line 12 - might be worth a word about the age of consent for sexual intercourse in general in many of the countries you are discussing here. In the UK that age is 16 years and the 15-19 age bracket obviously includes part of that period. I would argue this may help frame the introduction from that viewpoint.

This appears to reference a section of text which has since been deleted (based on another reviewer's comments). We trust that this deletion also addresses this reviewer's concern.

Methods

Starting at Line 40, I would argue this section should be in the introduction, I accept it is framing the sample but one might also argue it is addressing why you have chosen to undertake this study in this particular setting.

We acknowledge this comment - we have opted to keep the paragraph as is in the Method section. The introduction was trimmed (based on the first reviewer's comments), and the lighter introduction suits the flow of the paper nicely.

Line 48 - define SRH on first use

We have now spelled out 'sexual and reproductive health', thanks for catching this.

Table 1 - I would argue this is results, the demographics of the sample are a key aspect of what you found and how your sample was composed.

This concern is acknowledged. We've captured only the eligibility criteria in Table 1 and endeavoured to describe more of the demographics of our actual participants (insofar as it was appropriate for us to capture them – see our response to the first reviewer's comments on qualitative results presenting - in the Results section (in a section entitled 'qualitative method participant characteristics')

My general feeling is that the methods need to be re-written. I accept it is a nested part of a larger study but that should not prevent an accurate précis of what was done in this study. I am interested in the fact this is a mixed-methods study yet does not appear to demonstrate a link between the quant and qual aspects. How did the survey inform the focus of the qual (as would be the case in true mixed methods) or was this a case of a parallel quant and qual study?

The reviewer's concern is noted. Mixed Methods studies necessitate the inclusion of a qualitative and quantitative component. We refer to this excellent summary by Schoonenboom and Johnson, which captures the scope and nuance of what mixed methods studies can accomplish. Per the Schoonenboom and Johnson description, mixed methods studies can be conducted in parallel or sequentially, with regards to data collection. With regards to data analysis, mixed methods studies can be independent or dependent – the latter implies that data analysis for one method needs to be completed and informs the subsequent method.

*In our case, the data collection and analysis which led to this paper would be **parallel and independent**, as we use the qualitative and quantitative data to answer different research questions in a complementary way (what kinds of young people use pharmacies? – quantitative data; why are they appealing? – qualitative). The broader study from which this paper originates (which included five methods, one more – a mystery shopper exercise - than is presented here), however, would be best categorized as a parallel, dependent study. This all falls under the umbrella of 'mixed methods'. So, for the sake of keeping the methods parsimonious and clear, we've opted to keep the umbrella term: mixed methods study.*

P5 Line 4- how was a random set of pharmacies generated? Did you use a specific approach to randomisation? Once this is clarified there is also a need to clarify the location of the pharmacies, were the pharmacies chosen from a variety of neighbourhoods and therefore representative of different sociodemographic groups?

In response to this query, all mapped pharmacies went into an Excel database and we used the random number generator function to generate a list of random numbers associated with each pharmacy. We ordered the numbers (lowest to highest) and then sent out data collectors to work their way down the list (giving them ~5 pharmacies at a time, with check-ins in between to assess whether we were nearing saturation). There was no specific approach to randomisation; however, pharmacies from across the study area (described in paragraph 1 of the Methods section) were interviewed. We've added some brief clarifications to this end in the text (Line 131).

What theoretical approach was taken to the interviews? , What analytical framework was used? I can see a reference to data saturation in the text. There is very little in terms of how you have gone about triangulation of the qualitative data to manage bias?

This comment is appreciated. The theoretical framing which informed all qualitative data collection was based on WHO-defined dimensions of quality health services for adolescents. We have added text to this end and described how two of these dimensions influenced our qualitative data analysis. Regarding triangulation, we used the same codes to analyse qualitative data from all groups of participants: IDI participants with first-hand purchasing experience, adult pharmacy personnel with first-hand dispensing experience, and youth from the community who could speak to broader perceptions. This allowed us to verify – across different sources – salient themes. We particularly scrutinized cases where our pharmacy personnel and youth participants disagreed with each other. Due to space we have not included this level of detail in the method section – however, we do refer to it in our Limitations paragraph in the discussion.

Results

You use the term 'modern contraceptive' for the first time here. Im presuming thesis to differentiate between other methods of contraception. Might it be worth defining what you mean here clearly in the introduction ?

We appreciate this catch – we've added this clarification in the introduction, as suggested.

Characteristics - I find this difficult as I don't know which contraceptive products are available from non-medical outlets (including pharmacies) in Kenya. You have not framed that in the introduction. Are all of these products widely available from other sources or are they limited to pharmacies. The relationships are therefore more obvious if there are only medical outlets to use to obtain contraception.

This comment is noted and in response, we've added a clarification to the third paragraph of the Introduction.

In terms of the presentation of the qual data Im not clear what themes were identified in the thematic analysis. If the themes in the table are those I would argue some of these are essentially the same thing. What is the difference between service appeal and convenience ? Are these things not generally about access?

The reviewer's comment is noted with thanks – based on this and that of another reviewer, we have described the thematic analysis in further detail in the Methods section. Additionally, based on this comment, we've done some renaming and endeavoured to provide a BRIEF description of each of the major categories in the table (outlet appeal, personnel appeal, and service appeal) in Table 4 to clarify why these have been grouped as they are. We hope that this makes the service appeal/convenience distinction clearer – 'service' was narrowing in on the purchasing transaction itself, while convenience stayed with the location/opening hours of the physical premises.

I would argue the paper needs a conclusion

We appreciate this comment – we've revised a significant part of the discussion and the conclusion accordingly.

Reviewer: 3

Reviewer Name: Paula Tavrow

Institution and Country:

Fielding School of Public Health, UCLA

USA

Please state any competing interests or state 'None declared': None

Please leave your comments for the authors below See attachment with my comments
We have pasted the attached comments below

Comments on: “Mixed-methods study on pharmacies as contraception providers to Kenyan young people: who uses them and why?” by Paula Tavrow

This article makes an important contribution to the literature on adolescent contraceptive use in Africa—specifically where Kenya young people obtain their contraceptive methods and why so many rely on pharmacies, even where health facilities are free. I enjoyed reading it, particularly the comments

from the young people, which were well-selected and organized. Having been working in Kenya for the past two decades, I also appreciate that this topic needs more attention.

Our thanks for these kind words – we appreciated having a reviewer with this level of familiarity of Kenya.

The main weaknesses of the paper center on its imprecision, inattention to the implications of this reliance on pharmacies, and general readability. In my review, I will seek to highlight the areas of imprecision and pose some questions to be answered. Regarding readability, suffice to say that the authors frequently use compound sentences with colons or semi-colons, when simple declarative sentences would serve them better. The paper should be rewritten to remove colons and semi-colons except where strictly necessary.

This is noted with thanks and agreement. We have endeavoured to get rid of overly long sentences, and overuse of colons/semi-colons.

Also, the discussion is rambling and overlong. It should be broken into discreet topic areas with subheadings.

This comment is appreciated. We have reviewed and tightened the Discussion based on these comments and those of the other reviewers. The discussion follows the BMJ's instructions for authors and so we have withheld adding subheadings for the moment, though absolutely can do if needed.

Limitations should appear near the end, followed by conclusions.

The reviewer's concern is noted. We structured the Discussion according to the BMJ's Instructions for Authors, which calls for strengths/limitations of studies to be presented immediately after the principal findings. Likely reasons for BMJ preference are described in this 1999 Doherty & Smith editorial. Should the editor indicate that this no longer appropriate practice, we will of course change this.

Concerning imprecision, the authors frequently make statements that are not borne out by the paper's data or do not convey the issue accurately. Here are two prime examples from the abstract:

□ **“Public sector contraceptive services often struggle to meet the needs of young people around**

the world. Instead, private pharmacies have been demonstrated to be a relied-upon source of modern contraception for young people.” This is misleading. Public sector services are not “struggling” to meet the needs of young people. Instead, they are often very unfriendly to young people, as well as inconvenient, not confidential and not private. These are the main reasons that young people visit pharmacies instead. Pharmacies have not “been demonstrated” to be sources of contraception. They become the sources because young people find them more acceptable.

The reviewer's comment is noted. We have deleted the ‘Background’ section of the abstract entirely (as instructed, and in order to comply with BMJ abstract guidelines) but we have modified corresponding parts of the Introduction to address this.

□ **“Among surveyed participants, 59% had used contraception purchased from a pharmacy at last sex.”** This is inaccurate. It should read: “Among surveyed participants who had used a

modern form of contraception at last sex, 59% reported that they had purchased the method from a pharmacy.” The distinction is important. If one starts with “all surveyed participants (N=740),” one must conclude that 21% (153/740) reported that they had used contraception purchased from a pharmacy at last sex (based on Table 2). Also, the authors need to note that this is *as reported* by the young people. Those living at home may be reluctant to admit to ever having sex or ever using contraception.

Note that this “59%” figure also is the starting point of the Discussion. But, again, this is misleading. Again per the data in Table 2, *nearly half* of young people who are having sex are not using any contraception at all, mainly because of psycho-social barriers (including fears of side effects, fears of

others knowing they are sexually active, etc.). Of those who do use modern contraception, 59% are using methods purchased at pharmacies, which are the *least effective in preventing pregnancy in actual*

use (condoms and EC pills). More on this issue later.

We greatly appreciate the reviewer's comment – this was erroneously removed from the abstract in an earlier effort to comply with word count limitations. Our thanks for spotting this. We have corrected it. The point of those who are not at all using is also an important point and in the revised Discussion (from Line 380) we have endeavoured to place pharmacies (and other contraceptive points) in context. Improving services in these outlets is necessary but not sufficient to address contraceptive need of ALL young people (including non-users). We hope that this addresses the reviewer's concern.

Another way that the paper is imprecise is in Table 2. For baseline characteristics, it would be better to show the results by gender, particularly if the desire is to compare the results to the Kenya DHS, which

presents its data by gender. Even if there is a limitation in the data that it is not known who supplied the contraceptive method (for condoms), it still would be more useful to view the data by gender. Showing the data by gender would increase precision for the subsequent discussion. Also, I do not understand why the age data was bifurcated into “18-19” and “20-24”. I think that it would be best presented in thirds (e.g., 18-20, 21-22, 23-24) to see if rising age changes some of the contraceptive use dynamics.

We greatly appreciate this point. We have revised Table 2 accordingly to show baseline characteristics according to gender. We are preserving the age data split of 18-19 and 20-24 as a contribution to global calls to improve age disaggregation of adolescent-specific sexual and reproductive health data, for programmes, routine monitoring, and research. The period of adolescence ends at 19 so anyone conducting adolescent health-specific systematic reviews/meta-analyses will be able to use this data.

When initially analysing the data, we did subdivide the data further to see if our oldest participants had different contraceptive accessing behaviour, but did not find that this was the case. We attribute this to our very narrow age range and imagine that if our participants had comprised the full spectrum of UN-defined ‘youth’ (persons aged 15-24), we would have been more likely to see a significant difference between our younger and older young people.

It was curious to have a supplementary table that is not in the main text, when it was addressing one of the key study questions. I believe it would be best to have data presented as “retail (pharmacy/shop)” vs “non-retail (health centres, etc.), and to have the rows sum to 100% (not the columns). This would help us to see who uses retail vs. non-retail, as per the study question. In my view, separating out shops

from pharmacies isn't that useful, because some young people cannot tell the difference. The real question is: who uses retail rather than public/NGO clinics?

We absolutely concur with the reviewer's comments that retail vs non-retail is an extremely important question. However, we don't agree that young people – at least in our study area – can't distinguish between the two. In every one of our focus group discussions (where participants were explicitly asked to list sources of contraception), participants distinguished between pharmacies and shops. This may be because in Kenya (unlike most other countries in the region), pharmacies are the only legally-recognized retail drug outlet. There are no lower-level drug shops which blur the line between pharmacies and shops.

Like this reviewer, while running our quantitative analysis, we also considered collapsing pharmacies and shops. We, too, believe they share many similar qualities. We ultimately did not, as our qualitative methods had intentionally focused on pharmacies, rather than all retail outlets. As there are important distinctions between retail pharmacies (theoretically run by a licensed pharmacist) and retail shops (no professional requirements, one step removed from the health system) and how they are viewed by the community, we felt it unbalanced that paper to present quantitative data on retail outlets as a whole but qualitative data on pharmacies alone. We have added clarification about shops being out of our study's scope to the Discussion (Line 374)

Another area of imprecision surrounds the “relationship status.” How is “single” different from

“dating”? Since both are sexually active, does “single” signify that the last sex was a “one night stand” or “transactional sex”? This would be useful to differentiate. Are those in a “dating” relationship more likely to use clinics and hormonal methods?

The reviewer’s comment is noted with thanks. As this was a study featuring young people aged 18-24 – the typical ‘marital status’ demographics used (single, married/cohabiting, divorced, widowed) felt incomplete for capturing the pre-marital relationships that young people engage in. We therefore broadened (in consultation with our young data collectors and peer educators in the community) the list of options to better capture ‘relationship status’. We have updated our tables (Table 2 and 3) to describe these in full. Our survey also had a question about the relationship to the sexual partner. We had no participant who acknowledged sex either with a commercial sex worker, or with a client (as a commercial sex worker). Unfortunately, we had no questions which captured other forms of transactional sex.

Those who self-identified as ‘dating’ were most likely to get their contraception at last sex from a pharmacy compared with those of another relationship status. This was not apparent from how we previously presented the results, so we changed the relationship status reference group in our analysis to better highlight this (see Table 3)

Regarding implications, this is where the authors miss the boat in my opinion. Why are young people turning to pharmacies for their supplies? It is mainly owing to three reasons: health facilities are not youth-friendly, community norms are not accepting of sexually-active unmarried people, and comprehensive sexuality education in schools is generally non-existent. Hence, young, unmarried, nulliparous people seek the convenience and “privacy” (by which they mean the anonymity or not being seen by the community) of pharmacies to get those modern methods which they believe are least likely to be “harmful”. Without good information about implants, injectables, IUCDs, and oral contraceptives, the young people fall back on condoms and ECPs. Yet both have only 80-85% efficacy in real use. This leads to very high rates of unwanted pregnancies and illicit abortions—both of which surprisingly are unmentioned in the paper.

This is appreciated and based on this comment and this reviewer’s earlier comments, we have added a paragraph to the discussion focusing on the need for broader interventions which address the need for accurate information, addressing stigmas associated with use of contraception, agency to demand it, and broader barriers to accessing it – from any outlet. We have also noted the importance of being able to promote ‘highly effective forms of contraception’. However, we have not attempted to imply in this paper that young people should be switching away from condoms/ECP. In the word count provided, we cannot capture the complexities of negotiating safe sex within partnerships, or addressing broader ecological determinants of contraceptive uptake – important factors which may leave a young woman with no other choice than to use ECP. Additionally, condoms as a barrier method provide important STI/HIV protection and getting into dual protection is also not something we have space for under the word count.

For their discussion, the authors need to look not only at those who visit pharmacies, but also at those who do not, to get a clearer picture of what is happening. Their discussion needs to discuss this phenomenon. Namely, married young people, and/or those with children, are more likely to go to clinics because it is socially acceptable to do so. They are thus able to get more effective contraception.

On the other hand, nurses are averse to providing ECP to girls more than once. They feel it should only be for “emergency use” (even though it is actually fine to use frequently, just less effective). For girls who rely on ECP, they therefore need to get it from pharmacies.

We agree that this would be interesting to further explore – with word count limit we had to make a call on which variables to unpack. As having children and being married were not significant in the multivariate analysis, we opted to not to discuss them in detail. Instead we focus on the type of contraception used (the strongest correlation to choosing a pharmacy).

Per this suggestion, in the second paragraph, we have alluded to the need to possibly negotiate with health care workers for ECP. Looking into the literature based on this reviewer’s comments, we’ve also found emerging evidence from other countries in the region

indicating that public health facilities may opt not to stock ECP at all – we've also referenced this in the paragraph.

Lastly, the above realities should affect the conclusions of the authors. The pharmacy "solution" is not optimal for the reasons I laid out. Yes, it would be better if pharmacies could offer a more comprehensive range of methods. But the health services still do need to be improved and comprehensive sexuality education (that includes contraceptive methods) needs to be given in the schools, so that young people are aware of all of their options and do not need to rely mainly on pharmacies, which provide limited options for family planning. The current system is not only less affordable for young people (and may create greater health disparities since low income will have reduced access) but is also more prone to contraceptive failure since it relies on condoms and ECPs.

The ensuing social consequences--which may include unwanted births, dropping out of university, and botched abortions--can have dire long-term effects, particularly for girls

We, particularly with the revisions suggested by this reviewer, feel like we have addressed this through the Discussion revision. From Line 362, we have indicated that there needs to be a 'network' of providers (supported by broader ecosystem of interventions which increase information, demand, etc - see Line 385) to catch young people where they appear. Ideally, as we note in the revised text, strengthened services would also include referrals to providers who can offer highly effective forms of contraception. We hope these modifications are acceptable to the reviewer.

FORMATTING AMENDMENTS (if any)

Required amendments will be listed here; please include these changes in your revised version:

- Remove Background:

-On the Abstract section, please remove Background to comply with the Journal's structured abstract format.

We note this comment and have deleted this section accordingly.

- Please write a correct format of abstract for research paper (max. 300 words) including the following headings (please note that for RCTs there is a specific CONSORT extension for abstracts). Below is your guide:

The remaining subheadings are in line with the BMJ's suggested abstract headings. This is not presenting the results of an RCT so we have non included the CONSORT extension.

Objectives: clear statement of main study aim and major hypothesis/research question.

Setting: level of care e.g. primary, secondary; number of participating centres. Generalise; don't use the name of a specific centre, but give geographical location if important

Participants: numbers entering and completing the study; sex and ethnic group if appropriate. Clear definitions of selection, entry and exclusion criteria

Interventions (only relevant for interventional studies): what, how, when and how long (this can be deleted if there were no interventions)

Primary and secondary outcome measures: planned (i.e. in the protocol) and those finally measured (if different, explain why)

Results: main results with (for quantitative studies) 95% confidence intervals and, where appropriate, the exact level of statistical significance and the number need to treat/harm.

Whenever possible, state absolute rather than relative risks

Conclusions: primary conclusions and their implications, suggest areas for further research if appropriate. Do not go beyond the data in the article

Trial registration: registry and number (for clinical trials and, if available, for observational studies and systematic reviews).

VERSION 2 – REVIEW

REVIEWER	Efi Mantzourani Cardiff School of Pharmacy and Pharmaceutical Sciences, UK
REVIEW RETURNED	20-Feb-2020

GENERAL COMMENTS	Many thanks to the authors for taking the time to revise their manuscript, which is now significantly improved. I believe they have addressed the main points within the wordcount allowed.
---

REVIEWER	Andy Husband Newcastle University England
REVIEW RETURNED	14-Feb-2020

GENERAL COMMENTS	Thanks for the review of this paper and for the opportunity to read it again. The term 'last sex' is not something I have ever read before, you state that it is common parlance in this academic area. I would raise it as questionable syntax, but I don't intend to overly press the issue. My first comment on methods, still stands. The methods section is confused and unclear. The framing of the sample and the rationale for the study both need to be in the Introduction, in my opinion. Obviously the line numbers have changed, but the section that did begin on line 40 needs to be moved in my opinion. Table 1 is results not methods, I would argue this point has not been addressed. I will leave this to the discretion of the handling editor. I'm not sure the point that this is common practice in qualitative research does stand, certainly participants are described in the methods of qual papers, but not in the specific way this table reports, what are clearly, results. There is no clarification on the theoretical background to the qualitative section and no discussion of analytical frameworks. The answer to this point is explaining something different from what the question asks. There should be a description of the qualitative framework used and the analytical approach to the data. This is attributed to a lack of space in the response as is the point above in respect of the introduction. I would argue its about writing more concisely. The presentation of the resubmission is a little confusing, but unless I'm looking at the wrong document I can't see a conclusion or a limitations section as requested by other reviewers.
--

REVIEWER	Paula Tavrow UCLA Fielding School of Public Health USA
REVIEW RETURNED	31-Jan-2020

GENERAL COMMENTS	I was impressed with the comments and the care that the authors took to address all of the reviewers' concerns. I think that the paper is now readable, accurate, and useful. I believe it is ready for publication.
--

VERSION 2 – AUTHOR RESPONSE

Reviewer(s)' Comments to Author:

Reviewer: 3

Reviewer Name: Paula Tavrow

Institution and Country:

UCLA Fielding School of Public Health

USA

Please state any competing interests or state 'None declared': None declared

Please leave your comments for the authors below I was impressed with the comments and the care that the authors took to address all of the reviewers' concerns. I think that the paper is now readable, accurate, and useful. I believe it is ready for publication.

Our thanks to the reviewer for her comments – we are pleased that she is satisfied by our revisions.

Reviewer: 2

Reviewer Name: Andy Husband

Institution and Country:

Newcastle University

England

Please state any competing interests or state 'None declared': None declared

Please leave your comments for the authors below Thanks for the review of this paper and for the opportunity to read it again.

The term 'last sex' is not something I have ever read before, you state that it is common parlance in this academic area. I would raise it as questionable syntax, but I don't intend to overly press the issue.

Per this suggestion, we have changed 'at last sex' to 'at last sexual intercourse' wherever it occurs in the text.

My first comment on methods, still stands. The methods section is confused and unclear. The framing of the sample and the rationale for the study both need to be in the Introduction, in my opinion. Obviously the line numbers have changed, but the section that did begin on line 40 needs to be moved in my opinion.

We have moved the framing of the study location and sample to the introduction as requested – we hope that this adequately addresses the reviewer's concerns.

Table 1 is results not methods, I would argue this point has not been addressed. I will leave this to the discretion of the handling editor. I'm not sure the point that this is common practice in qualitative research does stand, certainly participants are described in the methods of qual papers, but not in the specific way this table reports, what are clearly, results.

We align with the suggestion of the reviewer to leave this decision to the discretion of the handling editor. We have reproduced our earlier response to this concern for the editor's ease of reference.

"It is standard practice in qualitative research to present the number of data collection events (interviews, FGDs, etc – as well as the numbers of participants who were involved) in the methods section rather than the results section. This is confirmed by the SRQR checklist as well. This is not the norm for quantitative studies. That said, and factoring in the changes proposed by reviewers to the results section, we feel the most straightforward and parsimonious way to present the data is to keep the table where it currently is. As a result we would like to keep the place on the number of participants as it stands and have not introduced specific changes"

There is no clarification on the theoretical background to the qualitative section and no discussion of analytical frameworks. The answer to this point is explaining something different from what the question asks. There should be a description of the qualitative framework used and the analytical approach to the data. This is attributed to a lack of space in the response as is the point above in respect of the introduction. I would argue its about writing more concisely.

We acknowledge the reviewer’s comment and have attempted to concisely address this in the Methods section. Grounded theory informed qualitative data collection – it allowed us to iteratively adapt question guides based on emerging themes. We have now explicitly mentioned this in the Data collection and Management subsection.

Data were analysed using the Framework Method, which allowed comparison of emerging themes across different groups of interviewees through inductive and deductive approaches. The second paragraph of the Analysis subsection presents key steps (transcription, familiarization with interview, coding, developing a framework, etc) in the same order as the key stages from this 2013 detailed description by Gale et al.

The presentation of the resubmission is a little confusing, but unless I'm looking at the wrong document I can't see a conclusion or a limitations section as requested by other reviewers.

This may have been an issue with the uploading of the resubmission in the system. The limitations are paragraph three of the discussion (keeping in line BMJ’s Instructions for Authors, which calls for strengths/limitations of studies to be presented immediately after the principal findings). The last paragraph (revised based on the first round of comments) is the conclusion –per submission guidelines, we did not label it as such, though we can do if the editor requires.

Reviewer: 1

Reviewer Name: Efi Mantzourani

Institution and Country: Cardiff School of Pharmacy and Pharmaceutical Sciences, UK Please state any competing interests or state ‘None declared’: None declared

Please leave your comments for the authors below Many thanks to the authors for taking the time to revise their manuscript, which is now significantly improved. I believe they have addressed the main points within the wordcount allowed.

Our thanks to the reviewer for her comments – we are happy that our revisions have addressed her suggestions.

VERSION 3 – REVIEW

REVIEWER	Andy Husband Newcastle University
REVIEW RETURNED	20-Mar-2020
GENERAL COMMENTS	I don't intend to continue with the discussion on Table 1, if the journalisms happy to publish this then so be it.